# Immune Checkpoint Inhibitors Targeting the PD-1/PD-L1 Pathway in Advanced, Recurrent Endometrial Cancer: A Scoping Review with SWOT Analysis

**DOI:** 10.3390/cancers15184632

**Published:** 2023-09-19

**Authors:** Racheal Louise Johnson, Subhasheenee Ganesan, Amudha Thangavelu, Georgios Theophilou, Diederick de Jong, Richard Hutson, David Nugent, Timothy Broadhead, Alexandros Laios, Michele Cummings, Nicolas Michel Orsi

**Affiliations:** 1Department of Gynaecological Oncology, St James’s University Hospital, Leeds LS9 7TF, UK; 2Leeds Institute of Medical Research, St James’s University Hospital, The University of Leeds, Leeds LS9 7TF, UK

**Keywords:** endometrial cancer, immunotherapy, PD-1, PD-L1, lenvatinib

## Abstract

**Simple Summary:**

This review will summarise the landmark clinical trials leading to the first tissue-agnostic approval of immune checkpoint inhibitors in specific molecular profiles of recurrent endometrial cancer (EC). As this treatment is a novel therapy and yet to be integrated into routine clinical use in the United Kingdom for EC patients, we will explore its strengths, including the ability to provide clinical survival benefit in patients with poor prognostic features, and its weaknesses, outlining immunotherapy toxicity and lack of availability for other molecular subgroups. We will define the opportunities this therapy presents, such as current trials investigating immunotherapy in combination with traditional therapy and/or novel targets, as well as threats to this treatment, such as financial implications and the practicalities of novel drug delivery and monitoring.

**Abstract:**

Results of recent clinical trials using the immune check point inhibitors (ICI) pembrolizumab or dostarlimab with/without lenvatinib has led to their approval for specific molecular subgroups of advanced recurrent endometrial cancer (EC). Herein, we summarise the clinical data leading to this first tissue-agnostic approval. As this novel therapy is not yet available in the United Kingdom standard care setting, we explore the strengths, weaknesses, opportunities, and threats (SWOT) of ICI treatment in EC. Major databases were searched focusing on clinical trials using programmed cell death protein 1 (PD-1) and its ligand (PD-L1) ICI which ultimately contributed to anti-PD-1 approval in EC. We performed a data quality assessment, reviewing survival and safety analysis. We included 15 studies involving 1609 EC patients: 458 with mismatch repair deficiency (MMRd)/microsatellite instability-high (MSI-H) status and 1084 with mismatch repair proficiency/microsatellite stable (MMRp/MSS) status. Pembrolizumab/dostarlimab have been approved for MMRd ECs, with the addition of lenvatinib for MMRp cases in the recurrent setting. Future efforts will focus on the pathological assessment of biomarkers to determine molecular phenotypes that correlate with response or resistance to ICI in order to identify patients most likely to benefit from this treatment.

## 1. Introduction

Endometrial cancer (EC) is the commonest gynaecological malignancy in the developed world, the fourth most common female cancer in the United Kingdom (UK), and its rising incidence is reflective of the global obesity pandemic [1]. Despite advances in rapid diagnosis and treatment, mortality rates have remained steady in the UK since the 1970s, with around 2400 annual deaths. The prognosis for advanced or recurrent EC still remains poor, with a 5-year survival of stage IV disease around 15% [2,3].

The cornerstone of EC treatment is surgical resection, which as a minimum includes a hysterectomy, with/without adjuvant chemotherapy, pelvic radiotherapy, or brachytherapy. The treatment strategy is tailored to individual risk factors such as age, stage, grade, histopathological subtype, depth of myometrial invasion, lymphovascular space invasion (LVSI), lymph node metastasis, and, more recently, molecular classification. Prior to the new molecular EC guidelines, histopathological subtype was crucial for stratifying adjuvant therapy.

Generally, adjuvant radiotherapy is the treatment of choice in advanced stage low-grade endometrioid EC, with the addition of chemotherapy considered for advanced stage, high-grade EC [4]. However, the histopathological grading of endometroid EC can be subjective, as reflected by comparatively low rates of inter-observer reproducibility across histopathologists [5,6]. Given that inaccurate histopathological diagnosis can result in suboptimal or over-treatment of patients, efforts have been made to develop a more robust classification system. In 2013, The Cancer Genome Atlas (TCGA) produced a comprehensive EC molecular classification based on four distinct genomic subgroups [7], each associated with the Proactive Molecular Risk Classifier for Endometrial Cancer (ProMisE) study prognostic data [8]. These included:(i)ultramutated EC defined by pathogenic mutations within the DNA polymerase epsilon (*POLE*) catalytic subunit, present in around 10% of endometrioid ECs;(ii)mismatch repair deficient (MMRd) EC, which corresponds to a microsatellite instability-high (MSI-H) phenotype and presents in around 40% of endometrioid and 2% of serous ECs;(iii)copy number low tumours exhibiting low somatic copy number alterations (SCNAs) associated with wild-type *TP53,* frequently associated with *PTEN, PIK3CA*, and *KRAS* mutations, and present in low-grade endometrioid EC; and(iv)copy number high tumours characterised by near universal *TP53* mutations and high SCNAs, most commonly occurring in serous EC.

The ProMisE study demonstrated the most favourable survival outcomes in the ultra-mutated *POLE* subtype and worst survival outcomes in women with *p53* abnormal (p53ab) tumours [8]. The most recent EC guidelines thus use molecular classification to determine the need for adjuvant treatment [4]. Notably, *POLE* mutant, ultra-mutated tumours have greater progression-free survival (PFS) when matched for stage, grade, and morphological subtype, and therefore do not require adjuvant treatment in early stage (I–II) disease [9,10]. Indeed, the Refining Adjuvant treatment In EC Based On molecular features (RAINBO) phase II trial is randomising patients with *POLE* mutated tumours to de-escalation of adjuvant treatment after surgery even for stage III tumours (NCT05255653) [11]. Patients with early stage Ia p53ab tumours (without LVSI) are classed as having an intermediate risk of recurrence and may benefit from adjuvant radiotherapy, whereas for those with advanced stage p53ab tumours or non-endometrioid histological subtype, combined radiotherapy and adjuvant chemotherapy is recommended. Patients with MMRd or no specific molecular profile tumours have an intermediate prognosis and adjuvant treatment is guided by additional clinical variables (e.g., LVSI and grade) [4]. However, treatment options are limited when chemotherapy resistance develops, while repeated courses of radiotherapy are associated with profound toxicity. There are currently no UK National Institute for Health and Care Excellence (NICE) recommended second-line systemic treatments for advanced EC in routine clinical practice. This unmet clinical need has thus motivated research for alternative therapies for these patients.

Throughout evolution, the immune system has been honed to maximise pathogen surveillance and eradication, whilst minimising damage to the host. This is mediated by its ability to differentiate between self and non-self antigens. This ability is facilitated by components of the innate immunity and immune checkpoints which regulate the magnitude of the adaptive immune response, and thus generate central tolerance [12]. In this context, major histocompatibility protein complexes (MHC) I and II are responsible for presenting peptide antigens on host cell surfaces to T-cells, to enable ‘self’ recognition, thereby preventing autoimmune responses. While CD8+ T cells recognise the immunogenic MHC-I peptides present on all nucleated cells, their CD4+ counterparts recognise MHC-II peptides on antigen-presenting cells (APCs; e.g., macrophages and dendritic cells), eliciting cytokine production and inducing effector T-cell differentiation. Analogously, tumour-associated antigen presentation to T-cells stimulates their proliferation and cytotoxic CD8+ T-cell killing of cancer cells, making them an attractive ally in the use of immunotherapy.

T-cells express immune checkpoints such as programmed death-1 (PD-1) and cytotoxic T lymphocyte antigen 4 (CTLA-4) which are induced on T-cell activation and act as a safeguard to regulate/suppress CD8+ T-cell cytotoxic function [12]. PD-1 expression on T-cells is upregulated following T-cell-receptor (TCR) engagement with MHC, resulting in overexpression of ligands PD-L1 and PD-L2 on APCs. Chronic engagement results in T-cell exhaustion via various mechanisms. For example, PD-1 contains two tyrosine motifs in its cytoplasmic tail, an immunoreceptor tyrosine-based inhibition motif, and an immunoreceptor tyrosine-based switch motif (ITSM). Motifs are phosphorylated upon PD-1 engagement with PD-L1, inducing recruitment of Src-homology 2 domain-containing phosphatase (SHP)-1 and SHP-2 to the cytoplasmic portion of PD-1. Recruitment of this complex is dependent on ITSM and exacerbates PD-1′s ability to block cytokine synthesis and limit T-cell expansion through downstream signalling [13]. In healthy tissues, these pathways serve to repress the activity of potentially autoreactive T-cells whilst ensuring a tailored pathogen response. However, cancer cells upregulate immune checkpoint molecules such as PD-L1 to evade detection and destruction by the immune system, thus enabling tumour survival and progression [14]. It is this blockade of negative feedback signalling to immune cells which underlies the *modus operandi* of ICIs.

The clinical application of ICIs to restore anti-cancer T-cell function in the TME has received United States Food and Drug Administration (FDA) approval for a variety of tumours with impressive, robust responses reported in malignant melanoma and lung cancer [15,16]. Inhibiting PD-1 signalling using monoclonal antibodies targeting PD-1 or its ligand, programmed death ligand 1 (PD-L1), has significantly improved survival outcomes in solid tumours characterised by MMRd or MSI-H status [17,18]. Thus, in May 2017, the FDA approved anti-PD-1 pembrolizumab therapy for any solid tumour failing prior systemic treatment that had MSI-H/MMRd molecular profiles, representing the first tissue-agnostic drug approval. The GARNET [19] and KEYNOTE-158 [18] clinical trials confirmed the survival benefit of anti-PD-1 monotherapy with dostarlimab and pembrolizumab, respectively, for MMRd/MSI-H ECs which had either progressed on prior treatment or were unresectable. This resulted in FDA approval for both agents and UK accessibility of dostarlimab specifically for this cohort in March 2022, heralding the dawn of a new era for immunotherapy in EC [20]. Response to anti-PD-1/PD-L1 monotherapy in EC patients with MMR proficient (MMRp) or microsatellite stable (MSS) profiles has been limited (overall response rate (ORR) 3–13%) [21,22]. However, opportunities for successful response to anti-PD-1 therapy in these patients were transformed by the addition of the multiple tyrosine kinase inhibitor lenvatinib. This was confirmed by the phase III randomised control trial (RCT) KEYNOTE-775, which demonstrated superior survival outcomes when combining pembrolizumab and lenvatinib compared to doxorubicin and paclitaxel chemotherapy for MMRp/MSS EC patients. This led to FDA approval for this combination in this population in July 2021 [23].

Although these drugs have achieved regulatory approval, their adoption in routine clinical practice is not fully embraced due to a lack of randomised control trial (RCT) evidence. Therefore, in the UK, dostarlimab is currently unavailable in the national health service (NHS) but via the cancer drug fund, and NICE’s recent appraisal of pembrolizumab and lenvatinib in June 2023 recommends this combination to EC patients who have progressed on or after platinum-based chemotherapy and are not fit for surgery or radiotherapy, only if companies provide them according to commercial arrangements [20,24,25]. Furthermore, although the evidence supporting a durable clinical benefit of ICIs is promising, drug resistance is also common, motivating the investigation of immunotherapy in combination with both traditional and novel therapies to sustain an anti-tumour effect [26]. Thus, in addition to our systematic review of clinical trials using PD-1/PD-L1 inhibitors in advanced, recurrent EC that lead to drug approval, we used a ‘SWOT’ framework [27,28] to analyse the strengths, weaknesses, opportunities, and threats for the potential mainstreaming of this new therapy for EC patients given that ICI is not available in the UK in the standard care setting.

## 2. Materials and Methods

This review was registered with PROSPERO (CRD42022372144). A systematic search of the PubMed/MEDLINE database using the subheadings “endometrial cancer” and “immunotherapy” was conducted specifically for clinical trials (Appendix A, Table A1). In addition, reference lists were reviewed from retrieved papers to identify potential additional relevant studies.

### 2.1. Inclusion Criteria

This systematic review was structured on ICI focused on anti-PD-1/PD-L1 therapy in EC of any histopathological type, molecular profile, or stage. Only studies reporting EC were covered, including mixed malignancy cohorts. Prospective and retrospective observational studies, RCTs, case-controlled studies, case-series, and abstracts were included. Inclusion criteria dictated that studies must have described at least one of the following primary outcome measures: overall response rate (ORR) assessed according to the Response Evaluation Criteria in Solid Tumours (RECIST) with computerised tomography (CT) or magnetic resonance imaging (MRI) per study protocol, survival data including progression-free survival (PFS) and overall survival (OS). Secondary outcome measures included therapeutic safety profile.

### 2.2. Exclusion Criteria

Excluded studies comprised those not published in English, those on immunotherapy agents other than PD-1/PD-L1 inhibitors (e.g., CTLA-4 inhibitors, T-cell therapy, anti-cancer vaccines, etc.), or those reporting on gynaecological malignancies not including EC.

### 2.3. Data Extraction

Two independent reviewers (R.L.J., S.G.) screened studies for inclusion. Data were extracted by both reviewers, and any conflict was resolved by the senior author.

### 2.4. Quality Assessment

The quality of included studies was evaluated for internal (bias specific to the study) and external (representativeness of outcome) validity based on Agency for Healthcare Research and Quality (AHRQ) guidelines [29]. For internal validity, we assessed the study design (prospective, retrospective) and evaluated bias across five domains, including patient selection, study performance (variation from protocol), attrition (adequacy of follow-up confirmed if >90%), detection (valid and reliably measured inclusion/exclusion criteria and interventions implemented consistently across all study participants), and reporting bias (pre-determined outcomes all reported). External validity was based on valid and reliable outcome reporting such as ORR, survival analysis, and safety data.

## 3. Results

The search algorithm yielded 2049 studies, which were initially screened, and based on the appropriateness of their title, *n* = 44 were retained. From this initial cohort, a total of 23 articles were selected for full-text review as they appeared to meet inclusion criteria from abstract screening. Of these, 13 were eligible for data extraction as they fulfilled the inclusion criteria. The reference lists from these 13 articles were screened and a further 2 texts were identified, such that 15 papers were included in the final analysis (Figure 1). This consisted of two retrospective single institution cohort analyses, three phase Ia/Ib trials, nine phase II trials, and one phase III RCT. Data were extracted on the ORR, survival, and safety of each trial.

The survival and safety data of included trials evaluating PD-1/PD-L1 pathway inhibitor in EC is summarised in Table 1. Six studies evaluated anti-PD-1 ICI monotherapy with pembrolizumab, with five recruiting patients with MMRd/MSI-H molecular EC subtypes [30,31,32,33,34] and one focussing on PD-L1 positive ECs [21]. These six studies included a total of 157 patients, of which 108 received pembrolizumab 200 mg intravenous (IV) every 3 weeks until progression/toxicity, and 49 received pembrolizumab 10 mg/kg intravenously every 2 weeks until progression/toxicity with an average ORR of 44.6%. Despite evidence suggesting that both fixed and weight-based pembrolizumab regimes are appropriate and with neither providing an advantage, a higher ORR was seen with the former regimen compared to the latter (48.6% versus 40.6%, respectively), paving the way for a fixed pembrolizumab dose receiving approval for patients with advanced, recurrent MMRd ECs who have progressed on systemic treatment and are not candidates for either surgery or radiotherapy [35]. In addition, the KEYNOTE-158 phase II study provided quality of life (QoL) data over 111 weeks. Amongst patients with complete or partial response (CR/PR), mean scores improved from baseline for pain, fatigue, insomnia, loss of appetite, and constipation, but remained stable for nausea and vomiting. This contrasted with those of patients with progressive disease who had exacerbated scores for nausea and vomiting but with other scores remaining stable [36]. One phase II study reviewed the alternative anti-PD-1 inhibitor nivolumab. This included 22 EC patients with an ORR of 22.7% for the entire cohort; only eight patients had molecular data with 100% ORR in the two MSI-H patients and 0% ORR in the six MSS patients [37]. Albeit in a small cohort, a lower overall ORR was observed in this study compared to the pembrolizumab studies [21,30,31,32,33,34].

The phase I GARNET trial reviewed the activity and safety of the anti-PD-1 agent dostarlimab for 129 MMRd/MSI-H and 161 MMRp/MSS EC patients, with an ORR of 47% versus 14.1%, respectively, with a median follow up of 16.3 months in the MMRd/MSI-H group. Median duration of response was not reached for either cohort. However, Kaplan–Meier survival estimates at 12 and 18 months between MMRd/MSI-H and MMRp/MSS cohorts were 90.9% versus 62.1% and 80.1% versus 62.1%, respectively [38]. This led to the FDA approval of this dosing regimen of dostarlimab for MMRd EC for patients who progress on, or progress following prior treatment with, platinum-based chemotherapy in April 2021. Dostarlimab is accessible to UK patients via the Cancer Drug fund until more evidence is reviewed by NICE [20].

PD-L1 inhibitors as monotherapy have also been evaluated in phase I/II trials, collectively including 117 patients with EC: 52 patients with MMRd/MSI-H, 63 with MMRp/MSS, and the remainder of unknown molecular status [22,39,40]. The mean ORR was significantly higher for MMRd/MSI-H versus MMRp/MSS EC (43.5% versus 4.6%). The largest of these trials directly compared 36 MMRd and 35 MMRp ECs treated with durvalumab and demonstrated a significant survival benefit in the MMRd versus MMRp cohorts, with a 1-year OS of 71% versus 51%, respectively. This was the only trial to include health-related (HR)-QoL data for anti-PD-L1 trials. Notably, 35% of patients’ HR-QoL increased from baseline at 3 months in the MMRd cohort compared to 8% in their MMRp counterparts. Improvements were seen in pain and fatigue QoL domains for MMRd compared to MMRp in 33% versus 9% and 28% versus 14% of patients, respectively [22].

Rates of successful response to anti-PD-1 in advanced, recurrent MMRp EC patients were significantly changed with the addition of lenvatinib. Three studies reviewed the combination of pembrolizumab (200 mg IV every 3 weeks until disease progression/toxicity) and lenvatinib (20 mg once a day orally). One phase Ib/II trial reported a moderate ORR with this combination of therapy of 37.2% in MMRp patients and 63.3% in MRRd [41]. Although this was significantly improved for MMRd patients, there was a greater ORR improvement in MMRp patients, where the reported ORR achieved by ICI monotherapy ranged from 0% to 14.1% [22,37,38,39]. The efficacy of this combination versus standard chemotherapy was confirmed by the randomised phase III KEYNOTE-775 trial of 827 advanced ECs (667 MMRp and 130 MMRd) randomised to pembrolizumab and lenvatinib versus chemotherapy, which demonstrated an ORR of 31.9% versus 14.7%, respectively, and significantly improved survival outcomes (PFS 7.2 versus 3.8 months; OS 18.3 versus 11.4 months, respectively) for the entire cohort. A similar effect was seen in the MMRp subgroup with an ORR 30.3% in the pembrolizumab and lenvatinib arm versus 15.1% with chemotherapy alone (PFS 6.6 versus 3.8 months; OS 17.4 versus 12 months, respectively) [23]. This secured FDA approval for this combination for EC patients with an MSS/MMRp profile in July 2021. Additionally, one study reviewed the dose of lenvatinib, comparing 20 mg versus 14 mg alongside pembrolizumab in 70 EC patients. While this had no impact on survival, there was a significant reduction of lenvatinib discontinuation at the lower versus the higher dose (36.8% versus 82.9%, respectively), challenging the merit of the currently approved regimen [42].

**Table 1 cancers-15-04632-t001:** Survival and safety data of clinical trials evaluating PD-1/PD-L1 pathway inhibitors in EC.

Study	Phase and Intervention	Patient Cohort	Response %	Survival Data (Median/ % Survival/HR)	Total TRAE any Grade/TRAE ≥Grade 3 (%)	Specific IRAE ≥Grade 3 (%)
Choi et al., 2020 [34]	Retrospective single institute Pembrolizumab 200 mg IV every 3 weeks until progression/toxicity	31 5 MMRd or MSI-H 45% ≥3 prior lines chemotherapy	ORR 40; CR 20; PR 20	PFS 2.5 months; OS 14.3 months	64.5/9.6	9.6 (1 patient death of new onset interstitial lung disease)
Lui et al., 2019 [40]	Phase Ia Atezolizumab 15 mg/kg IV every 3 weeks until progression/toxicity	15 1 MSI-H, 12 MSS, 3 unknown status 53.3% ≥2 prior lines chemotherapy 66.7% prior radiation	ORR 13.3 (2 patients); CR 0; PR 13.3 1 responder MSI-H, 1 unknown, both had ≥ 5% PD-L1 expression	PFS 1.4 months; OS 9.6 months DOR in responders 7.3 and 16.3 months	46.7/20	13.4
Oakin et al., The GARNET Trial, 2020 [38]	Phase I Dostarlimab 500 mg IV every 3 weeks for 4 doses, then 1000 mg every 6 weeks until disease progression	264 108 MMRd/MSI-H and 156 MMRp/MSS 11% ≥3 prior lines chemotherapy 65% prior radiation	MMRd/MSI-H ORR 43.4; CR 10.4; PR 33 MMRp/MSS ORR 14.1; CR 1.9; PR 12.1	KM estimates 1-year OS MMRd 90.9% vs. MMRp 62.1%	67.6/16.6	7.6
Ott et al., The KEYNOTE-028 Trial., 2017 [21]	Phase Ib Pembrolizumab 10 mg/kg every 2 weeks until progression/toxicity	24 PD-L1 positive 1 MSI-H, 18 MSS 41.6% ≥3 prior lines chemotherapy	ORR 13; CR 0; PR 13; SD 13	1-year PFS 14.3%; 1-year OS 53% Median OS and PFS NR	54.2/16.7	8.3
Tamura et al., 2019 [37]	Phase II Nivolumab 240 mg IV every 2 weeks until disease progression/toxicity	22 2 MSI-H, 6 MSS 17% ≥3 prior lines chemotherapy 17% prior radiation	ORR 22.7 MSI-H 100, MSS 0	PFS 3.2 months; OS 8.7 months PFS MSI-H NRPFS MSS 2.2 months	61/17	8
O’Malley et al., The KEYNOTE-158 Trial., 2020 [33]	Phase II Pembrolizumab 200 mg IV every 3 weeks for 2 years or until progression	79 All MSI-H/MMRd 28% ≥3 prior lines chemotherapy 71% prior radiation	ORR 48; CR 14; PR 34	PFS 13.1 months; median OS NR KM estimates 1-year OS 88%	76/12	7
Konstantinopoulos et al., 2019 [39]	Phase II Avelumab 10 mg/kg IV every 2 weeks until progression/toxicity	31 15 MMRd 16 MMRp 41.9% ≥3 prior lines chemotherapy	MMRd ORR 26.7; CR 6.6; PR 20 MMRp ORR 6.2; CR 0; PR 6.25	PFS 4.4 months; median OS NR	71/19.4	12.9
Antill et al., 2021 [22]	Phase II Durvalumab 1500 mg IV every 4 weeks until progression/toxicity	71 35 MMRp and 36 MMRd progressed after ≥ 1 line of therapy 42% ≥2 prior lines chemotherapy 66% prior radiation	MMRd: ORR 47; CR 16.6; PR 30.5 MMRp: ORR 3; CR 0; PR 3	MMRd PFS 8.3 months; median OS NR; 1-year OS 71% MMRp PFS 1.8 months; OS 12 months; 1-year OS 51%	21/3 19.7 Improvement to QoL: MMRd 25; MMRp 9 Improvement of pain: MMRd 33; MMRp 10	3
Bellone et al., 2021 [32]	Phase II Pembrolizumab 200 mg IV every 3 weeks until progression/toxicity	24 MSI-H 6 harboured Lynch-like MMRd and 18 sporadic (MLH1 promoter methylation) Median 1 prior line systemic therapy range 1–5)	ORR 58 (100 in Lynch-like; 44 in sporadic)	3-year PFS 100% Lynch-like and 30% sporadic	Not reported/6.8	Not reported
Fader et al., 2016 [31] Abstract	Phase II Pembrolizumab 10 mg/kg IV every 2 weeks	9 MMRd failed ≥2 previous therapies	ORR 56; CR 11.1; PR 44.4	1-year OS 89%, median OS NR	Not reported	Not reported
Le et al., 2017 [30]	Phase II Pembrolizumab 10 mg/kg IV every 2 weeks	86 MMRd multiple tumours 15 MMRd EC 47% ≥3 prior lines chemotherapy for all tumour types	EC ORR 53; CR 20; PR 33	PFS 18.1 months Median OS NR all tumour types	74/26 all tumour types	21 across all tumour types
Makker et al., 2020 [41]	Phase Ib/II Lenvatinib 20 mg OD + pembrolizumab 200 mg IV every 3 weeks	108 11 MSI-H/MMRd; 94 MSS/MMRp, 3 no molecular data 37% ≥2 prior lines chemotherapy	Total cohort: ORR 38.9; CR 7.4; PR 31.5 MSI-H/MMRd: ORR 63.6 MSS/MMRp: ORR 37.2	MSI-H/MMRd: PFS 7.4 months; median OS NR MSS/MMRp: PFS 7.4 months; OS 16.4 months	97.2/69.4	40.3 (2 treatment related deaths of sepsis and intracranial haemorrhage)
Taylor et al., 2020 [43]	Phase Ib/II Lenvatinib 20 mg OD + pembrolizumab 200 mg IV every 3 weeks	23 ECs (molecular status unknown) 74% ≥2 prior lines chemotherapy	ORR 52; CR 9; PR 44	PFS 9.7 months	97/67	62
How et al., 2021 [42]	Single-institution, retrospective cohort study Lenvatinib (14 mg or 20 mg) + pembrolizumab	70 1 MSI-H, 69 MSS 16 = 20 mg 54 = 14 mg Median lines prior systemic therapy 2 (range 1–9)	Lenvatinib 20 mg: ORR 28.6; CR 0; PR 28.6 Lenvatinib 14 mg: ORR 38.2; CR 4.3; PR 34 No significant difference	Lenvatinib 20 mg: PFS 3.2 months; OS 8.6 months Lenvatinib 14 mg: PFS 5.5 months; OS 9.4 months	Not reported. 32.9% hospitalisation due to TRAE Treatment discontinued due to lenvatinib dose: 82.9% (20 mg) and 38.6% (14 mg)	Not reported
Makker et al., The KEYNOTE-775 Trial, 2022 [23]	Phase III RCT Lenvatinib 20 mg OD + pembrolizumab 200 mg IV every 3 weeks (LP) versus CT doxorubicin and paclitaxel	827 667 MMRp 130 MMRd 77.5% prior 1 line chemotherapy 43.5% prior radiation	Lenvatinib + pembrolizumab vs. CT; ORR total cohort 31.9 vs. 14.7 MMRp: 30.3 vs. 15.5 MMRd: 40.0 vs. 12.0	Lenvatinib + pembrolizumab vs. chemotherapy Total cohort: PFS 7.2 vs. 3.8 months (HR 0.56) MMRp: PFS 6.6 vs. 3.8 months (HR 0.60)	Lenvatinib + pembrolizumab 99.8/88.9 Chemotherapy 99.5/72.7 No difference in long term QoL scores	44.5 Not applicable

Abbreviations; endometrial cancer (EC), mismatch repair proficient (MMRp), mismatch repair deficient (MMRd), microsatellite instability high (MSI-H), microsatellite stable (MSS), phase of trial (Ph), high-grade (HG), chemotherapy (CT), overall response rate (ORR), complete response (CR), partial response (PR), progression free survival (PFS), overall survival (OS), duration of response (DOR), hazard ratio (HR), stable disease (SD), median survival not reached (NR), once a day (OD), intravenously (IV), treatment-related adverse events (TRAE), immune-related adverse events (IRAE), grade 3 (G3), quality of life (QoL).

### Quality Analysis

The majority of the trials included herein (87%) were non-blinded prospective cohort analyses, with only one RCT. This creates uncertainty surrounding the clinical survival benefits of ICI as it only includes one study with a direct comparison with other treatments. In addition, although it is well documented that MMRd/MSI-H tumours respond to ICI therapy, only one third of these studies included a direct comparison (of ≥16 patients) with MMRp/MSS molecular status, limiting the ability to ascertain the comparative survival benefits between different EC molecular subgroups [22,23,38,39,41]. Moreover, there are limited long-term data, with KEYNOTE-028 and KEYNOTE-158 trials only providing 1-year survival data on a combined population of 103 EC patients [21,33] and Bellone et al. providing 3-year survival data on only 49 EC patients [32]. However, there were clear inclusion/exclusion criteria to reduce selection bias which ensured that appropriate patients were recruited in most studies (93%). It was recognised that prior exposure to immunotherapy could bias the performance of the intended treatment, and this was part of all studies’ exclusion criteria. All studies maintained fidelity to the intervention protocol. Adequate follow-up was demonstrated in all instances, with clear explanations justifying participant removal from the trial (usually due to toxicity and/or death). Three studies (20%) recruited 100 or more patients, with a mean study population of 40 participants across the remainder. Detection bias evaluated the difference in duration of follow-up between groups, whether interventions were implemented consistently across all study participants in prospective studies, and if outcomes were assessed using valid and reliable measures. None of the studies were blinded. All studies reported pre-specified outcomes including ORR and survival outcomes (PFS/OS). Most studies (86%) reported treatment-related adverse events and included specific reports of immune-related events (Figure 2).

## 4. SWOT Analysis of ICI in EC Patients

### 4.1. Strengths

The studies reviewed included recurrent advanced, metastatic, or relapsed disease often treated with at least one prior line of systemic therapy. For progression or relapse within 6 months of prior chemotherapy, response rates of second line chemotherapy are disappointing, hormone therapy provides a limited survival benefit for those eligible, while repeat courses of radiotherapy are associated with profound toxicity [44,45,46]. Therefore, the ability of ICI to achieve a response in cohorts with such poor prognostic features (including the KEYNOTE-755 study’s superior survival outcomes compared to conventional therapy) highlights its therapeutic potential [23]. This offers hope to a cohort of patients for whom there have been no prior recommended second-line systemic treatments. Fader and co-workers reported an 89% 1-year OS with pembrolizumab in patients with 1–4 previous chemotherapy regimens [31]. Makker et al. demonstrated a durable response (30% patients with PFS of over 6 months) to pembrolizumab and lenvatinib in a cohort which included patients who had received two lines of prior chemotherapy [41]. Other case series have demonstrated PFS of 24 and 28 months in two relapsed, recurrent EC patients in response to pembrolizumab after chemoradiation [47]. These examples demonstrate the durability of ICI in sustaining an anti-tumour immune effect and translating it into sustainable clinical/survival benefits. Furthermore, safety data report that ICI monotherapy or in combination with lenvatinib is relatively well tolerated and is not associated with myelosuppression and subsequent sepsis, which is a well-recognised side-effect of chemotherapy. Additionally, QoL scores were not significantly different with pembrolizumab and lenvatinib compared to conventional chemotherapy in the KEYNOTE-775 RCT [23]. The clinical success of such therapies has motivated pharmaceutical companies to create ICI available as an ‘off the shelf’ intravenous preparation, making it more accessible in the clinical setting.

### 4.2. Weaknesses

Although ICI is often well tolerated, autoimmune toxicity can affect any organ and may require treatment discontinuation. While serious events (grade 3 or above) of pneumonitis and cardiotoxicity occur only in under 1% of patients treated with anti-PD-1/PD-L1 therapy, they can be fatal. Dermatologic toxicity and endocrinopathies are the most common immunotherapy-related toxicities, with the latter often being irreversible. Moreover, progressive neuropathies (e.g., Guillain–Barré syndrome), neuromuscular syndromes (e.g., myasthenia gravis), and aseptic meningitis/encephalitis can cause highly morbid and even life-threatening complications, which mandates close clinical monitoring [48]. Furthermore, the management of immune-related toxicity differs from that of chemotherapy. In the former, ICI treatment is abandoned, and corticosteroids are administered, while in the latter, a dose reduction can allow patients to continue anti-cancer therapy.

Frustratingly, the success of ICI in cancer patients as a whole is limited, with only an estimated 13% of patients eligible for, and responding to, ICI therapy [49]. Indeed, only patients with tumours harbouring certain molecular and immunological profiles exhibit a survival benefit with ICI, suggesting that immunotherapies need to be tailored to discrete molecular subtypes and individual TME immunogenic landscapes. MMR status currently determines eligibility for immunotherapy in EC. However, a predictive biomarker to identify ICI monotherapy responders or those benefiting from combined therapies, while avoiding the toxic side-effects of futile therapies, remains elusive. In the context of pembrolizumab, previous pan-cancer studies have demonstrated that PD-L1 expression and high tumour mutational burden (TMB) are associated with improved ORR [50,51]. A recent study of 366 patients treated with atezolizumab across different malignancies (non-small cell lung, renal cell, and urothelial carcinomas) evaluated PD-L1 status, TMB, and transcriptional profiling to identify predictive biomarkers of response and resistance to ICI. Unfortunately, multiple machine learning models deployed in this context failed to identify a unifying transcriptional signature predictive of ORR. While PD-L1 expression and high TMB correlated with increased ORR, the low specificity of these biomarkers limited their ability to accurately predict atezolizumab responders. Moreover, 10% of responders were PD-L1 negative with low TMB, suggesting that there may be independent or multifactorial mechanisms that contribute to treatment response. Importantly, significant molecular heterogeneity was observed across tumours, suggesting that multiple factors are likely at play in determining ICI response, which highlights the difficulty of biomarker-based patient stratification in this context. In this regard, a deep learning algorithm (Ensemble Learning for Immunotherapeutic Response Evaluation; ELISE) used estimates of intra-tumoural stroma and immune cell infiltration from gene set-enrichment analysis [52]. This performed with 100% area under the curve (AUC) in the test cohort and 99% in the validation cohort when predicting responses to atezolizumab in 76 patients with oesophageal adenocarcinoma [53]. More specifically, feature selection identified 442 RNAs that were significantly associated with atezolizumab response as well as others that were significantly downregulated in non-responders. The reduction of certain RNAs in non-responders correlated principally with dysregulated immune signalling (e.g., immunoglobulin production, antigen binding). The ELISE algorithm was extended to 79 patients with metastatic urothelial carcinoma to predict response to anti-PD-1/PD-L1 therapy, where it had an AUC of 88.8% [53]. These promising results suggest that large molecular datasets may have some merit in profiling individual TME landscapes to determine ICI response across a range of malignancies rather than relying on single biomarker assessments. However, multi-marker panels are both cost and resource intensive and, as such, largely remain confined to the academic setting.

While much emphasis has been placed on patient hyporesponsiveness to ICI, unexpected, accelerated tumour progression following the initiation of ICI therapy (termed hyperprogression) has been reported. Another phenomenon where initial tumour progression is followed by a clinical objective response is instead known as pseudoprogression. Both reactions are challenging for clinical practice since it is unclear which patients may benefit from continued treatment beyond the initial progression, and which warrant early interruption of therapy. Hyperprogression has no consensus definition per se, but may be described as a ≥2-fold increase in tumour size within a two-month period of commencing treatment. It is also associated with markedly worse survival. This pattern of hyperprogression has been reported in a number of retrospective studies [54,55,56,57,58]. A subset of 131 patients with 21 types of cancer who received anti-PD-1 therapy (including two patients with EC) suggested that such events were associated with older age (>65 years) [59]. Molecular profiling of another six patients (with urothelial carcinoma, lung adenocarcinoma, endometrial sarcoma, and triple-negative breast carcinoma) whose tumours displayed hyperprogression in response to anti-PD-1 and anti-CTLA-4 therapy appeared to be associated with MDM2/MDM4 amplifications (all cases) and EGFR mutations (two patients) [60]. Although MDM2/MDM4 amplification is more a feature of endometrial sarcoma rather than carcinoma, it underscores the fact that an underlying predisposing genomic profile may account for hyperprogression. Plausibly, however, disease progression may have occurred in such cases regardless of ICI therapy. In this regard, a post hoc analysis of two RCTs reviewing a total of 599 patients with small cell lung or gastric carcinomas on ICI (nivolumab, ipilimumab, or both) versus 290 patients on placebo showed no significant difference in tumour diameter on baseline and treatment evaluation CT scan, suggesting that reports of hyperprogression could simply reflect the natural course of the disease in certain patients rather than reflecting ICI-mediated progression [61].

### 4.3. Opportunities

#### 4.3.1. ICI Combined with Approved Cancer Therapy

Previous studies have demonstrated an association of MMRd/MSI-H colorectal cancer with significantly improved prognosis in response to combined correlation could apply to EC [62,63]. Cytotoxic chemotherapy induces direct cancer autophagy, causing the release of immunostimulatory molecules such as lysosomal ATP which promotes dendritic cell (DC) recruitment to the TME. In turn, DCs facilitate antigen presentation to T-cells to stimulate a cytotoxic, anti-tumour cellular reaction.

Furthermore, chemotherapy-induced cancer cell DNA damage leads to the accumulation of aberrant nucleic acids in the cytosol as well as their release from dying cancer cells. These activate the cGAS-STING (cyclic GMP-AMP stimulator of interferon genes) pathway and stimulate toll-like receptor (TLR 3 or 9) signalling, respectively, resulting in an increased production of Type I interferon (IFN) and the induction of innate immune defence. Together, these mechanisms help to promote DC-mediated tumour antigen presentation to CD8+ T-cells, potentially eliminating residual cancer cells [64]. Moreover, lymphodepletion following chemotherapy mediates an acute state of lymphopenia-induced T-cell proliferation, thereby increasing tumour-infiltrating lymphocyte populations available for ICI stimulation [65]. In this regard, agents such as cyclophosphamide have been shown to deplete tumour-infiltrating Treg numbers and their immunosuppressive function in mouse models [66]. This compelling body of evidence continues to encourage review of the use of ICI in combination with chemotherapy. Phase III trials are under way to review the effect of such combinations in EC cohorts (NCT03914612, NCT03603184) [67,68].

Increased tumour burden also positively correlates with reduced PD-1 immunotherapy efficacy in melanoma patients [69]. Fractionated radiation therapy can stimulate antitumour immunity and has the additional advantage of providing tumour debulking. Preclinical evidence demonstrates an increased intratumoural CD8+ T-cell infiltration post-radiation, indicating a potential clinical synergy with immunotherapy [70]. In this respect, early phase trials of melanoma and lung cancer treated with ICI and radiotherapy demonstrated prolonged survival with tolerable side-effects [71,72,73]. A phase I trial combining weekly bladder irradiation (36 Gy in 6 weekly fractions) with pembrolizumab in patients with metastatic or recurrent urothelial carcinoma was prematurely halted as the first five (out of six) patients experienced dose-limiting toxicity (including one rectal perforation). As this toxicity is greater than expected, the authors advised caution when combining high-dose pelvic radiotherapy and ICI [74]. The standard external beam pelvic radiotherapy dose for EC patients is usually 45 Gy in 6 weekly fractions with an estimated 5% rate of grade 3 adverse toxicity [75]. A cohort of 73 patients with advanced metastatic solid tumours that had progressed on standard therapy (including six with EC) received multi-site stereotactic body radiotherapy followed by pembrolizumab. This technique uses a multi-beam approach to target high-dose radiation to the tumour and limit toxicity to surrounding tissues. The ORR was a modest 13.2% but was accompanied by a 9.7% rate of dose-limiting toxicity, with six patients exhibiting severe treatment-related toxicity. In this setting, significantly increased levels of IFN-γ and granzyme K expression were noted in irradiated tumour biopsies compared to their pre-irradiated biopsy counterparts [76]. The potential merit of combined ICI and radiotherapy is the focus of a phase III, randomised, double-blind study of pembrolizumab versus placebo in combination with adjuvant chemotherapy with or without external beam radiotherapy (45 Gy with variable frequency) in high-risk EC patients post-surgical debulking (NCT04634877) [77]. Results are anticipated in 2025.

ICI and poly (ADP-ribose) polymerase (PARP) inhibitor (PARPi) combination therapy is also under investigation. PARP is involved in the repair of single-strand DNA breaks through the base excision repair pathway. In this context, PARPis lead to trapping of PARP proteins at sites of single-strand breaks, allowing them to persist unrepaired during DNA replication, thereby causing the accumulation of double-strand DNA breaks. Since BRCA1 and BRCA2 proteins participate in repairing DNA double-strand breaks via homologous recombination repair (HRR), the accumulation of DNA damage induced by PARPis selectively kills BRCA mutated/silenced cells [78]. Data have shown that PARPis are beneficial in cancers with a defect in DNA HRR, regardless of BRCA mutational status. Recent preclinical data in OC suggest that PARPis also act through diverse mechanisms to modulate the immune TME, providing a rationale for PARPi/ICI combination therapy [79]. For example, a novel PARP1/2 inhibitor (BMN 673) exhibited immunoregulatory effects in a BRCA1 mutated murine ovarian cancer model. BMN 673 treatment correlated with significantly increased numbers of natural killer (NK) cells and CD8+ T-cells, as well as IFN-γ production by the latter. BMN 673 inhibited cell proliferation and induced apoptosis in BRCA mutated cells, indicating that it may be a promising adjunct to immunotherapy [80]. This has translated clinically into early phase ovarian cancer trials combining PARPis with ICI, with one reporting an ORR of 25% regardless of BRCA mutational status using pembrolizumab, and another of 63% in relapsed BRCA1/2 mutated ovarian cancer using durvalumab [81,82]. Despite EC not being considered a hereditary component of BRCA mutation syndrome, it can nevertheless occur in such patients. Moreover, there is a higher incidence of EC in BRCA mutated patients compared to the baseline population [83]. Interestingly, 20–25% of ECs exhibit *TP53* mutations and *PTEN* mutations are frequent in type I EC’s, both of which are indirectly involved in HRR [7,84,85]. Homologous recombination deficiency (HRD) is associated with poor PFS in EC and predicts EC sensitivity to PARPi therapy in orthoptic murine models [86]. In vitro studies have indicated that PTEN-deficient EC cell lines demonstrate an increased sensitivity to PARPis compared to PTEN-intact cells. This association was reinforced by observations from a PTEN-deficient EC xenograft mouse model, which exhibited a significant reduction in tumour volume in response to PARPi treatment [87,88]. In this regard, a phase I/II trial of 31 advanced, pre-treated EC patients combining PARPi with metformin and cyclophosphamide demonstrated 61.5% non-progressive disease at 10 weeks with 5.1 month PFS (similar to the PFS observed with chemotherapy or immunotherapy in the KEYNOTE-775 trial, which covered advanced, recurrent, and pre-treated EC) [89]. Another phase II trial evaluated the merit of durvalumab and olaparib combination therapy in 50 patients with advanced, pre-treated, recurrent EC (14% carcinosarcomas, and 38% serous, 32% endometrioid, and 12% clear cell carcinomas, of which 59% were p53ab, 20% MMRd, and 20% had a non-specific molecular profile), and demonstrated a 34% 6-month PFS rate. Therapy was well tolerated but failed to meet their predefined 50% 6-month PFS target [90]. Current phase III clinical trials combining ICI and chemotherapy with or without PARPis for advanced recurrent EC are ongoing (Table 2).

As demonstrated by the addition of lenvatinib to pembrolizumab, combination therapy presents future opportunities for EC treatment. Lenvatinib is a multi-receptor tyrosine kinase inhibitor (TKI) shown to have immunomodulatory effects. When comparing the immune profiles of murine models of hepatocellular carcinoma, colon carcinoma and melanoma, lenvatinib-treated tumours showed increased CD8+ T-cell and plasmocytoid DC infiltrates and a reduction in tumour-associated macrophages (TAMs) compared to non-treated tumours. This anti-tumour immune response was enhanced by the addition of anti-PD-1 antibody which resulted in a greater frequency of IFN-γ secreting CD8+ T-cells and a further reduction in TAMs compared to lenvatinib monotherapy [91]. In this regard, a recent phase II study reviewed the efficacy of nivolumab monotherapy compared to combination with the receptor TKI inhibitor cobazantinib for advanced, recurrent EC in both ICI-naïve patients and those who had progressed on ICI monotherapy. PFS was improved in the combination arm of ICI-naïve patients by a median 5.3 months compared to monotherapy (median 1.9 months), with ORRs of 25% versus 11%, respectively. However, combination therapy was associated with a 64% rate of adverse events ≥grade 3 compared to only 6% with nivolumab monotherapy. In the post-ICI subgroup, rechallenge with nivolumab and cobazantinib achieved an ORR of 25% with a median duration of stable disease of 5.5 months in which, of note, nearly half of the re-challenged patients had carcinosarcoma [92]. In this respect, a phase Ib/II trial recruiting recurrent carcinosarcoma patients to receive cobazantinib and dostarlimab was registered in September 2022 (NCT05559879) [93].

Receptor tyrosine kinase signalling pathways also play a crucial role in angiogenesis, with drugs such as lenvatinib and cobazantinib targeting vascular endothelial growth factor (VEGF) receptors (VEGFRs) 1–3. The VEGF family contains key neoangiogenic regulators whose stimulation promotes endothelial survival, migration, and permeability [94]. Overexpression of VEGFs has been demonstrated in EC and is associated with poor survival [95]. In the TME, abnormal tumour vascularisation favours an immunosuppressive environment characterised by hypoxia and an allied low pH. In particular, VEGF-A promotes Treg proliferation [96] and enhanced PD-1 expression on CD8+ T-cells in vitro comparing samples from naïve versus colorectal tumour-bearing mice, providing a rationale for combining antiangiogenic treatment with ICI [97]. Recent results of the phase II EndoBARR trial evaluated the anti-PD-L1 agent atezolizumab, with bevacuzimab and rucaparib (PARPi) triple therapy in 30 recurrent EC patients previously treated with one or more lines of therapy. Median follow-up was 14.9 months with 1 (4%) patient exhibiting a complete response, 9 (39%) a partial response, and 13 (57%) having stable disease. Grade 3/4 events were noted in 50% of patients. Overall PFS was 5.3 months, with an enhanced clinical response noted in MMRd patients. Long term survival data are awaited [98]. Furthermore, a phase I study of recurrent gynaecological cancers (including one MSS EC) demonstrated a reduction in tumour volume in response to durvalumab (anti-PD-L1), olaparib (PARPi) and cediranib (VEGFR tyrosine kinase inhibitor) triple therapy [99]. Early phase trials recruiting women with advanced recurrent EC for atezolizumab (anti-PD-L1) and bevacizumab (anti-VEGF) combination therapy are ongoing (NCT03526432; Table 3) [100].

#### 4.3.2. Dual ICI Therapy

Nivolumab and ipilimumab (which targets CTLA-4) combination therapy was approved for advanced melanoma after demonstrating superior survival outcomes compared to ipilimumab monotherapy (63.5% versus 53.6% 2-year OS, respectively), albeit with greater treatment-related grade 3/4 adverse events (55% versus 20%, respectively) [101]. A phase II randomised trial of recurrent EC patients previously treated with platinum-based chemotherapy received durvalumab with or without tremelimumab (an anti-CTLA-4 agent). Results showed a modest ORR (ORR 14.8 versus 11.2%, respectively) and PFS (7.6 weeks versus 8.1 weeks, respectively) but with no significant improvement. Notably, there was a marked increase in treatment-related grade 3 events (32 versus 7%, respectively) [102]. A phase II randomised trial is currently recruiting MMRd recurrent EC patients to receive nivolumab with or without ipilimumab (NCT05112601) [103].

#### 4.3.3. ICI Combinations with Novel Therapies

The strategy of combining ICI with other targeted therapies is also under investigation. One actively pursued target is the cell surface protein folate receptor (FR) α, which is overexpressed in both ovarian and ECs, but restricted in healthy tissues [104,105]. This differential expression makes FRα an attractive tumour-associated antigen target for antibody-drug conjugates (ADC). In this regard, mirvetuximab soravtansine is an ADC comprising an FRα-binding antibody and the potent tubulin-targeting agent DM4. A phase III RCT assigned mirvetuximab soravtansine or standard chemotherapy to 366 patients with pre-treated advanced ovarian cancer. While differences in the study’s primary endpoint (PFS) did not reach statistical significance, ORR was significantly improved and CA-125 levels were significantly lower in patients with FRα positive tumours in the mirvetuximab soravtansine group [106]. Another phase III RCT replicating this approach in 430 pre-treated, advanced ovarian cancer patients is due to be completed later this year (NCT04209855) [107]. The activity of mirvetuximab soravtansine has been studied in high-grade endometrioid and serous ECs in the preclinical setting. Increased cytotoxicity was observed in response to mirvetuximab soravtansine in EC cell lines overexpressing FRα compared both to controls and those with low FRα expression. Moreover, in a xenograft mouse model of FRα overexpressing endometrioid EC, treatment with mirvetuximab soravtansine resulted in tumour regression and increased survival [108]. In addition to its direct cytotoxic effects, mirvetuximab soravtansine has been shown to activate peripheral blood monocytes and induce DC-mediated phagocytosis through Fc-FcγR interaction [109]. This offers a mechanistic rationale for combining mirvetuximab soravtansine with ICI and, in this regard, a phase II trial investigating this in FRα positive MMRp EC is ongoing (NCT03835819) [110].

Another phase Ib/II multi-cohort study is currently recruiting patients to evaluate the anti-PD-L1 agent atezolizumab in combination with targeted therapy based on tumour-specific genomic profiles in recurrent or persistent EC. Patients with no specific molecular signatures will receive atezolizumab and bevacizumab, while those with ≥16% genomic loss of heterozygosity (LOH) which is associated with HRD will be assigned to atezolizumab and the PARPi talazoparib. Patients with MSI-H tumours will receive atezolizumab and tiragolumab (NCT04486352) [111]. Tiragolumab is a monoclonal antibody against T-cell immunoreceptors with Ig and immunoreceptor tyrosine-based inhibitory motif domains (TIGIT), a co-inhibitory immune checkpoint receptor expressed on immune cells and upregulated on T-cells and NK cells in multiple solid tumour types [112]. Both TIGIT and its ligand poliovirus receptor-related 2 (PVRL2) are overexpressed in EC [113]. Blockade of TIGIT can enhance anti-PD-1/PD-L1 therapy, leading to improved ORR and PFS, as demonstrated by the CITYSCAPE phase II trial in non-small cell lung cancer (NSCLC). This led to FDA approval of this combination in PD-L1 positive NSCLC in 2021 [114].

The final molecular group specified in this trial includes patients with PI3K, protein kinase b (AKT), or PTEN-mutated tumours who will receive atezolizumab and the AKT inhibitor ipatasertib [111]. PI3K signalling leads to downstream activation of AKT and mechanistic target of rapamycin (mTOR) [115]. This intracellular signalling pathway is responsible for regulating processes as diverse as the cell cycle, metabolism, and angiogenesis [116]. Given that PTEN is an inhibitor of this pathway, it limits cell proliferation when acting as a tumour suppressor. The PI3K/AKT/mTOR pathway is characterised by extensive feedback loops and crosstalk with RAS and downstream RAF-MEK-ERK signalling [117]. A phase II trial reviewing the effectiveness of a MEK1/2 inhibitor (AZD6244, ARRY-142886) in recurrent or persistent EC failed to meet pre-trial expectations in terms of clinical efficacy, although it exhibited favourable tolerability [118]. While approval of PI3K inhibitors for the treatment of follicular lymphoma and mTOR inhibitors for renal cell carcinoma has been granted by the FDA, monotherapy targeting this pathway has only achieved a modest clinical effect in EC, such that no targeted therapies are currently available [119,120,121]. While the frequent mutations reported in PI3K/AKT and RAS/RAF pathways justified exploring the relative merits of dual target treatment, AKT inhibitor and MEK inhibitor combination therapy featured high levels of toxicity and did not demonstrate clinical efficacy in a phase Ib trial [122]. Nevertheless, PI3K/AKT signalling has been linked to activation of oestrogen-induced PD-L1 expression in oestrogen receptor-α positive EC cell lines. Oestrogen-induced PD-L1 expression is immunologically relevant since this is correlated with decreased IFN-γ and interleukin-2 production (which are both essential for T-cell function) in EC cell lines. Treatment with an AKT and PI3K inhibitor in vitro has been shown to block oestrogen-dependent PD-L1 expression, providing preclinical evidence to support combining atezolizumab with an AKT inhibitor [123]. Additionally, patients with unresectable advanced triple negative breast carcinoma received AKT inhibitor (IPAT), atezolizumab, and paclitaxel chemotherapy which demonstrated a 54% ORR and 7.2 months median PFS, highlighting synergy with ICI independent of oestrogen, leading to enrolment of this triplet to a phase III RCT whose results are awaited (NCT04177108) [124].

Novel approaches to ICI drug combination therapies have also involved the repurposing of existing drugs. For example, the drug ataluren is used to treat genetic disorders (e.g., Duchenne muscular dystrophy) and works by allowing the ribosome to read through premature stop codons (PTCs), thus inhibiting nonsense-mediated mRNA decay (NMD) and allowing the production of full length polypeptides [125]. As MMRd/MSI-H tumours have abundant frameshift mutations and, consequently, PTCs, ataluren is hypothesised to allow translation of out-of-frame code downstream of PTCs, leading to increased tumour neoantigen production, as evidenced by preclinical studies using NMD inhibitors on MSI-H CRCs [126]. In turn, this could synergise with immunotherapy and, in this vein, ataluren will be combined with pembrolizumab in MMRd EC (NCT04014530). Table 3 illustrates early phase EC trials combining ICI with alternative targeted therapies.

### 4.4. Threats

Despite ICI being off-the-shelf immunotherapy available for both MMRd/MSI-H and MMRp/MSS ECs, it is more costly than chemotherapy [127]. Moreover, the need for regular intravenous infusions of ICI dictates the need for increased healthcare service access as well as requiring specialist multidisciplinary team training (oncology nurses, doctors, and radiologists), both in terms of administering ICI to the appropriate patient and in the management of immune-related complications. Since ICI is only available for EC that has undergone MMR classification, its use requires a priori immunohistochemical evaluation of MMR status and, by inference, access to well-resourced histopathology services. In order to streamline EC molecular classification in the UK, the British Association of Gynaecological Pathologists (BAGP) created an algorithm in 2022 which proposed that all EC endometrial biopsies, regardless of histological type, should be tested for MMR, p53, and oestrogen receptor status by immunohistochemistry. *POLE* testing is now also available in the UK via the national genomic service, but it requires additional resources in terms of sample transport, processing, analysis, and reporting such that it is only recommended by the BAGP if it will alter clinical management [128].

Both the molecular classification of EC and delivery of ICI pose a financial burden. The NICE evidence review group evaluation indicated that the incremental cost-effectiveness ratio (ICER) is 49,454 GBP per quality-adjusted life year (QALY) gained with dostarlimab compared to current conventional treatment in EC [20]. Although NICE has never identified an ICER above which intervention should not be recommended, an ICER below 20,000 GBP per QALY gained is generally considered to be cost-effective [129]. By way of reference, a recent NICE cost-effectiveness review of adjuvant pembrolizumab therapy for completely resected stage 3 melanoma determined an ICER of 26,493 GBP per QALY gained. Immunotherapy for melanoma was appraised by NICE in 2015 suggesting that, in time, the ICER in EC will reduce, making treatment more cost-effective as it becomes more established [130]. Furthermore, despite also requiring considerable expertise and resources, the development of more novel immunotherapeutic approaches (e.g., cancer vaccines) may provide a more robust T-cell-redirecting anti-tumour effect which may eventually overshadow ICI [131].

## 5. Conclusions

In summary, the SWOT analysis revealed that the strengths of ICI revolved around improved patient survival, even in those with poor prognostic features, with more positive QoL profiles and tolerable side-effects compared to standard chemotherapy. Weaknesses pertained to ICI’s efficacy being limited to select populations as well as its toxicity, the need for treatment discontinuation if it arises, and its putative role in hyper-/pseudoprogression. Opportunities centred on exploiting ICI’s efficacy as part of combination therapy with either existing or novel agents. Finally, threats focused on the financial and healthcare infrastructure costs of ICI’s use, as well as its longevity in the face of novel immunotherapies (Figure 3).

ICI’s promise to offer a durable survival benefit for patients with advanced, recurrent EC hinges on its ability to reinvigorate the anti-cancer immune response. The clinical efficacy of ICI monotherapy is most evident in EC patients with an MMRd/MSI-H subtype; the addition of lenvatinib allows this survival benefit to extend to those without a typical ‘immunogenic’ profile. Importantly, these treatments have a good safety profile where reducing lenvatinib dose can further improve toxicity profile without compromising clinical benefit and/or long-term QoL compared to conventional therapies [23,42]. The results anticipated from ongoing RCTs aim to provide more robust evidence of durable survival benefits when compared with standard clinical treatment, which should drive a paradigm shift towards making immunotherapy available in routine clinical practice [24].

Historically, patient selection for adjuvant anticancer treatment such as chemotherapy was based on histopathological tumour subtypes (e.g., the addition of chemotherapy for advanced serous EC). However, ICI has been shown to have clinical effects across a variety of tumours (and their subtypes), including melanoma, renal cell carcinoma, non-small cell lung cancer, and EC, but with only a subset of patients exhibiting durable ICI monotherapy response. The ability to identify patients likely to benefit clinically from ICI would also contribute to making the therapy more cost-effective. Unfortunately, cost remains a significant drawback to ICI’s wider adoption. Future refinements in ICI regimen dosing and duration may ease this: for example, NICE recognised that a 6-weekly rather than 2-weekly pembrolizumab regime for advanced melanoma patients can reduce administrative burden and cost to the NHS without compromising clinical efficacy [130]. In addition, adaptive ICI dosing based on early interim radiographic assessment in melanoma patients found that those with a radiological response who then ceased further ICI therapy still had a survival benefit compared to standard treatment. This reduces the risk of toxicity whilst also presenting a cost-saving strategy [132]. Furthermore, the delivery of subcutaneous ICI to patients with metastatic solid tumours (including EC) has not compromised clinical efficacy compared to intravenous preparations [133,134]. The former approach may thus represent a more cost-effective, convenient delivery method, potentially even allowing patients to self-administer at home.

Finally, to achieve the maximum clinical benefit from immunotherapy, it is necessary to overcome tumour host immunotolerance. Clinical trials targeting multiple cancer pathways will establish the potential clinical benefits of both traditional and novel treatments when combined with ICI and shed light on the most effective approach to gain safe and durable clinical benefits in the context of EC. Taken together, the accrual of clinical immunotherapy experience and the evidence of sustained survival across a range of malignancies supported by advances in scientific research will encourage healthcare providers to alter service infrastructure to enable immunotherapy to be more readily available in the clinical setting.

## Figures and Tables

**Figure 1 cancers-15-04632-f001:**
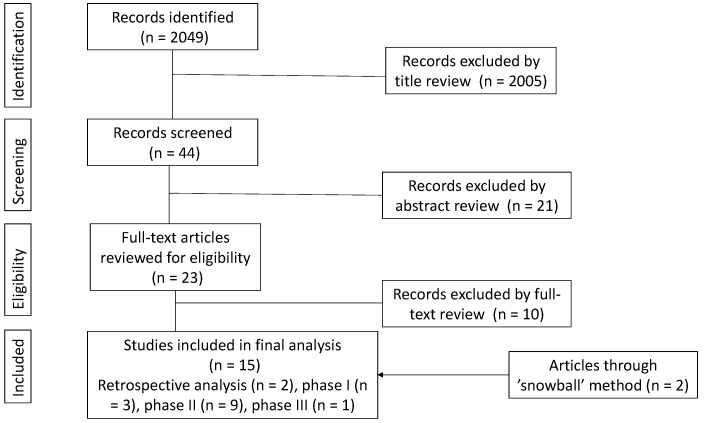
Search strategy flow diagram.

**Figure 2 cancers-15-04632-f002:**
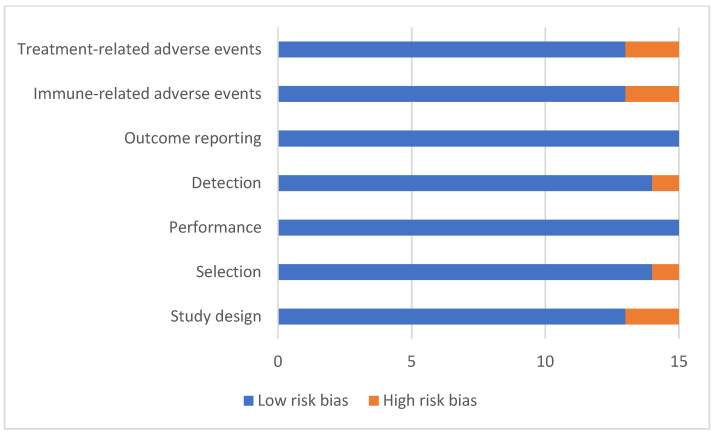
Quality assessment of included studies.

**Figure 3 cancers-15-04632-f003:**
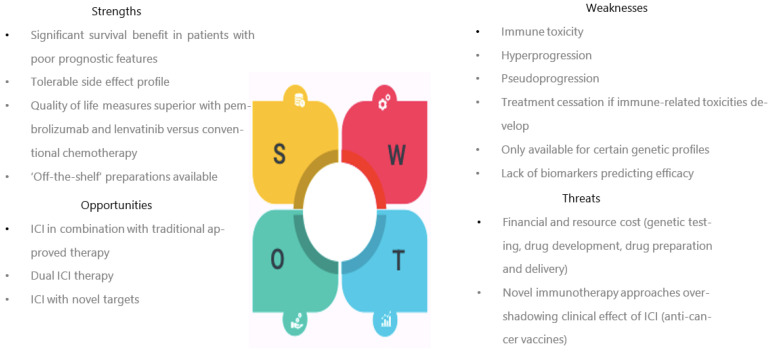
Immune checkpoint inhibitors targeting the PD-1:PD-L1 pathway in advanced, recurrent EC: summary of strengths, weaknesses, opportunities, and threats (SWOT).

**Table 2 cancers-15-04632-t002:** Phase III immunotherapy trials in combination with chemotherapy and/or PARPis for advanced or recurrent EC.

NCT Number	Description	No. Patients	Study Status	Study Name
NCT03914612	CPP or CP + pembrolizumab	590 MMRp 220 MMRd	Recruiting	NRG GY018
NCT03981796	CPP or CP + dostarlimab +/− niraparib or placebo	740	Active, recruitment period completed	RUBY
NCT03603184	CPP or CP + atezolizumab	550	Active, recruitment period completed	AtTEND
NCT04269200	CPP or durvalumab +/− olaparib maintenance or placebo	699	Recruiting	DUO-E
NCT05201547	CPP or CP + dostarlimab	142 MMRD	Recruiting	DOMENCIA

Abbreviations; carboplatin and paclitaxel (CP), carboplatin, paclitaxel, and placebo (CPP), United States National library of National Clinical Trials (NCT), mismatch repair proficient (MMRp), mismatch repair deficient (MMRd).

**Table 3 cancers-15-04632-t003:** Early phase trials of EC immunotherapy combinations.

NCT Number	Phase	Description	No. Patients	Inclusion	Study Name
NCT03932409 Recruiting	Ib	Pembrolizumab + vaginal cuff brachytherapy + chemotherapy	40	High-risk EC	FIERCE
NCT04014530 Recruiting	Exp	Pembrolizumab + ataluren	47	MMRd metastatic EC	ATAPEMBRO
NCT03835819 Recruiting	II	Pembrolizumab + mirvetuximab soravtansine	35	MSS/MMRp recurrent EC	-
NCT03526432 Active, not recruiting	II	Atezolizumab + bevacizumab	55	Advanced (III/IV) or recurrent EC	-
NCT04486352 Recruiting	Ib/II	Atezolizumab + bevacizumab or ipatasertib (AKT inhibitor) or talazoparib (PARPi) or tiragolumab (anti-TIGIT)	100	Advanced (III/IV) or recurrent EC PIK3CA/AKT1/PTEN-altered tumours for ipatasertib cohort	EndoMAP
NCT04444193 Unknown	Exp	Durvalumab + lenvatinib	20	Advanced (III/IV) or recurrent EC	DULECT-2020-2
NCT03660826 Recruiting	II	Durvalumab + olaparib or durvalumab + cediranib maleate (VEGFR-2 kinase inhibitor) or cediranib maleate + olaparib or as single agents	120	Recurrent or refractory EC	-
NCT03015129 Active, not recruiting	II	Durvalumab monotherapy or durvalumab + tremelimumab (anti-CTLA-4)	80	Recurrent EC	-
NCT05112601 Recruiting	II	Ipilimumab (anti-CTLA-4) + nivolumab	12	MMRd recurrent EC	-
NCT03367741 Active, not recruiting	II	Nivolumab monotherapy or nivolumab + cabozantinib (tyrosine kinase inhibitor)	84	Advanced (III/IV) or recurrent EC	-

Abbreviations; carboplatin and paclitaxel (CP, carboplatin, paclitaxel, and placebo (CPP), United States National library of National Clinical Trials (NCT), vascular endothelial growth factor receptor 2 (VEGFR-2), mismatch repair proficient (MMRp), mismatch repair deficient (MMRd), microsatellite stable (MSS) EC.

## Data Availability

Not applicable.

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
