# Peer review of "Immune Checkpoint Inhibitors Targeting the PD-1/PD-L1 Pathway in Advanced, Recurrent Endometrial Cancer: A Scoping Review with SWOT Analysis"

_cancers, 2023, doi:10.3390/cancers15184632_

Round 1

Reviewer 1 Report

Abstract should be clear and more concise. In abstract abbreviation need to be defined immune checkpoints inhibitors (ICI). Abstract  conclusion should be specific with recommendation for the  therapy, specifically for Immune checkpoint inhibitors targeting the PD-1/PD-L1 for  recurrent endometrial cancer    

Elaborate more in details on  strengths, weaknesses, opportunities, and threats (SWOT) approach

Elaborate more  on   Immune checkpoint inhibitors targeting the PD-1/PD-L1

Elaborate more  on  clinics  and histopathology of the recurrent endometrial cancer and options in treatment including chemotherapy, immunotherapy and correlate to this finding by the  SWOT analysis

Author Response

The authors would like to thank the Reviewers for reading this manuscript and for their suggestions for improvement.Please also find attached a copy of our revised manuscript with tracked changes.

Reviewer #1:

  1. Abstract should be clear and more concise. In abstract abbreviation need to be definedimmune checkpoints inhibitors (ICI). Abstract conclusion should be specific with recommendation for the therapy, specifically for Immune checkpoint inhibitors targeting the PD-1/PD-L1 for recurrent endometrial cancer.  

This has been elaborated on within the manuscript (please see amended abstract). Moreover, the requested recommendation has been included to read: “Pembrolizumab has been approved for MMRd ECs, with the supplementation of lenvatinib for MMRp cases in the recurrent setting” in the body of the text.

  1. Elaborate more in details on strengths, weaknesses, opportunities, and threats (SWOT) approach.

As ICI is not available in the UK in a standard mainstream care setting, the SWOT analysis is used as a framework to provide an objective assessment of ICI in clinical practice as this is a novel and emerging therapy in the UK. It defines the potential obstacles/threats of ICI in practice and discussed future opportunities that ICI may have, specifically for this cohort of patients. We have also included references of other SWOT analysis in a healthcare setting. The text has been amended to read: “Thus, in addition to our systematic review of clinical trials using PD-1/PD-L1 inhibitors in advanced, recurrent EC that lead to drug approval, we used a ‘SWOT’ framework [28,29] to analyse the strengths, weaknesses, opportunities, and threats for the potential mainstreaming of this new therapy for EC patients given that ICI is not available in the UK in the standard care setting” (see lines 513-517). Two appropriate supporting references have further been added:

Rallis KS et al, T-cell-based Immunotherapies for Haematological Cancers, Part A: A SWOT Analysis of Immune Checkpoint Inhibitors (ICIs) and Bispecific T-Cell Engagers (BiTEs). Anticancer Res. 2021 Mar;41(3):1123-1141. doi: 10.21873/anticanres.14870. PMID: 33788704.

Espinós JJ et al, Impact of chronic endometritis in infertility: a SWOT analysis. Reprod Biomed Online. 2021 May;42(5):939-951. doi: 10.1016/j.rbmo.2021.02.003. Epub 2021 Feb 11. PMID: 33736994.

  1. Elaborate more on Immune checkpoint inhibitors targeting the PD-1/PD-L1.

This has been done, as requested. The amended text now reads: “Throughout evolution, the immune system has been honed to maximise pathogen surveillance and eradication, whilst minimising damage to the host. This is mediated by its ability to differentiate between self and non-self antigens. This ability is facilitated by components of the innate immunity and immune checkpoints which regulate the mag-nitude of the adaptive immune response, and thus generate central tolerance.[12] In this context, major histocompatibility protein complexes (MHC) I and II are responsible for presenting peptide antigens on host cell surfaces to T-cells, to enable ‘self’ recognition thereby preventing autoimmune responses. While CD8+ T cells recognise the immu-nogenic MHC-I peptides present on all nucleated cells, their CD4+ counterparts recognise MHC-II peptides on antigen-presenting cells (APCs; e.g. macrophages and dendritic cells), eliciting cytokine production and inducing effector T-cell differentiation. Analo-gously, tumour-associated antigen presentation to T-cells stimulates their proliferation and cytotoxic CD8+ T-cell killing of cancer cells, making them an attractive ally in the use of immunotherapy.

T-cells express immune checkpoints such as programmed death-1 (PD-1) and cyto-toxic T lymphocyte antigen 4 (CTLA-4) which are induced on T-cell activation and act as a safeguard to regulate/suppress CD8+ T-cell cytotoxic function.[12] PD-1 expression on T-cells is upregulated following T-cell-receptor (TCR) engagement with MHC, resulting in overexpression of ligands PD-L1 and PD-L2 on APCs. Chronic engagement results in T-cell exhaustion via various mechanisms. For example, PD-1 contains two tyrosine motifs in its cytoplasmic tail, an immunoreceptor tyrosine-based inhibition motif and an im-munoreceptor tyrosine-based switch motif (ITSM). Motifs are phosphorylated upon PD-1 engagement with PD-L1, inducing recruitment of Src-homology 2 domain-containing phosphatase (SHP)-1 and SHP-2 to the cytoplasmic portion of PD-1. Recruitment of this complex is dependent on ITSM and exacerbates PD-1’s ability to block cytokine synthesis and limit T-cell expansion through downstream signalling.[13] In healthy tissues, these pathways serve to repress the activity of potentially autoreactive T-cells whilst ensuring a tailored pathogen response. However, cancer cells upregulate immune checkpoint molecules such as PD-L1 to evade detection and destruction by the immune system, thus enabling tumour survival and progression.[14] It is this blockade of negative feedback signalling to immune cells which underlies the modus operandi of ICIs” (lines 153-183).

  1. Elaborate more on  clinics  and histopathology of the recurrent endometrial cancer and options in treatment including chemotherapy, immunotherapy and correlate to this finding by the  SWOT analysis.

In terms of this point, there is a lack of approved second line systemic treatments for advanced/recurrent EC with evidence of clear survival benefit. As a result, they are frequently managed using a palliative approach (see lines 147-152), hence ICI being approved from such small studies.

Reviewer 2 Report

This study is interesting with clinical significance. Immunotherapy therapy has revolutionized the tumor therapy, especially in endometrial cancer. The authors put forward a new and comprehensive point of view on of immune checkpoint inhibitors targeting the PD-1/PD-L1 pathway in endometrial cancer. The followings are some comments to the authors.

Comments:

1.In order to better show IRAE, I suggest that IRAE G3 % should be added in the Table 1.

2.Whether the included studies had the same number of prior regimens. I suggest that number of prior regimens should be added in the Table 1. Because this is a very an important factor in the efficacy data.

3. How to balance the difference of MSI-H/MMRd ratio in 15 studies? Since the MSI-H/MMRd population can get better efficacy treated by immune checkpoint inhibitors.

4.How to balance the difference of dose in 15 studies?

Author Response

The authors would like to thank the Reviewers for reading this manuscript and for their suggestions for improvement.Please also find attached a copy of our revised manuscript with tracked changes.

Reviewer #2:

This study is interesting with clinical significance. Immunotherapy therapy has revolutionized the tumor therapy, especially in endometrial cancer. The authors put forward a new and comprehensive point of view on of immune checkpoint inhibitors targeting the PD-1/PD-L1 pathway in endometrial cancer. The followings are some comments to the authors.

Comments:

  1. In order to better show IRAE, I suggest that IRAE G3 % should be added in the Table 1. 

2. Whether the included studies had the same number of prior regimens. I suggest that number of prior regimens should be added in the Table 1. Because this is a very an important factor in the efficacy data.

This has been done, as requested (please see amended Table 1). Please note that many studies report only TRAE with a list of adverse reactions. We have added a column for specific IRAE ³ Grade 3 as listed by each study.

3. How to balance the difference of MSI-H/MMRd ratio in 15 studies? Since the MSI-H/MMRd population can get better efficacy treated by immune checkpoint inhibitors.

We agree that this is a valid point. It is well documented that MMRd/MSI-H patients have an excellent response to ICI – as a result, few studies include direct comparisons to other EC molecular subgroups, and this limits the ability to ascertain survival benefit in other EC molecular cohorts. This has been added to the quality analysis (see lines 732-735).

4. How to balance the difference of dose in 15 studies?

This is a valid point. We have provided an additional sentence in the results section (see lines 393-396) which underscores the fact that previous research has suggested that both fixed and weight-based pembrolizumab regimens are appropriate, with neither providing a clear survival advantage. However, since a higher ORR was demonstrated in these EC patients treated with a fixed dose, this management strategy was ultimately approved. In addition, a supportive refernce has been added:

Freshwater, T.; Kondic, A.; Ahamadi, M.; Li, C.H.; de Greef, R.; de Alwis, D.; Stone, J.A. Evaluation of dosing strategy for pembrolizumab for oncology indications. J. Immunother. Cancer 2017, 5, 1–9, doi:10.1186/S40425-017-0242-5/TABLES/2.